# Monoclonal Antibodies Targeting an *Opisthorchis viverrini* Extracellular Vesicle Tetraspanin Protect Hamsters against Challenge Infection

**DOI:** 10.3390/vaccines9070740

**Published:** 2021-07-05

**Authors:** Wuttipong Phumrattanaprapin, Mark Pearson, Darren Pickering, Bemnet Tedla, Michael Smout, Sujittra Chaiyadet, Paul J. Brindley, Alex Loukas, Thewarach Laha

**Affiliations:** 1Department of Parasitology, Faculty of Medicine, Khon Kaen University, Khon Kaen 40002, Thailand; wuttipongp@kkumail.com; 2Faculty of Veterinary Medicine and Applied Zoology, HRH Princess Chulabhorn College of Medical Science, Chulabhorn Royal Academy, Bangkok 10210, Thailand; 3Centre for Molecular Therapeutics, Australian Institute of Tropical Health and Medicine, James Cook University, Cairns, QLD 4878, Australia; mark.pearson@jcu.edu.au (M.P.); darren.pickering@jcu.edu.au (D.P.); bemnetamare.tedla@jcu.edu.au (B.T.); michael.smout@jcu.edu.au (M.S.); 4Tropical Medicine Graduate Program, Academic Affairs, Faculty of Medicine, Khon Kaen University, Khon Kaen 40002, Thailand; sujitch@kku.ac.th; 5Immunology and Tropical Medicine, Research Center for Neglected Diseases of Poverty, Department of Microbiology, School of Medicine & Health Sciences, George Washington University, Washington, DC 20037, USA; pbrindley@gwu.edu

**Keywords:** passive immunization, monoclonal antibody, *Opisthorchis viverrini*, tetraspanin

## Abstract

*Opisthorchis viverrini* causes severe pathology in the bile ducts of infected human hosts, and chronic infection can culminate in bile duct cancer. The prevention of infection by vaccination would decrease opisthorchiasis-induced morbidity and mortality. The tetraspanin protein, *Ov*-TSP-2, is located on the membrane of secreted extracellular vesicles (EVs), and is a candidate antigen for inclusion in a subunit vaccine. To address the role of anti-*Ov*-TSP-2 antibodies in protection, we assessed the protective capacity of anti-*Ov*-TSP-2 monoclonal antibodies (mAbs) against opisthorchiasis. Two anti-TSP-2 IgM mAbs, 1D6 and 3F5, and an isotype control were passively transferred to hamsters, followed by parasite challenge one day later. Hamsters that received 3F5 had 74.5% fewer adult flukes and 67.4% fewer eggs per gram of feces compared to hamsters that received the control IgM. Both 1D6 and 3F5 (but not the control IgM) blocked the uptake of fluke EVs by human bile duct epithelial cells in vitro. This is the first report of passive immunization against human liver fluke infection, and the findings portend the feasibility of antibody-directed therapies for liver fluke infection, bolstering the selection of TSPs as components of a subunit vaccine for opisthorchiasis and fluke infections generally.

## 1. Introduction

The human liver fluke, *Opisthorchis viverrini*, resides in the bile ducts within the biliary tree, where it establishes chronic infection that drives long-term inflammation, hepatobiliary damage, and bile duct obstruction, leading to cholangiocarcinoma (CCA) [1]. Humans become infected by ingesting the infective stage metacercariae from raw or undercooked cyprinoid fish dishes [2,3]. We recently showed that subunit vaccines targeting tetraspanins (TSPs) on the surface of fluke-secreted extracellular vesicles (EVs) confer protection in a hamster challenge model of *O. viverrini* infection, and that the antibodies against the TSPs present on the EV surface block the uptake of fluke EVs by the host bile duct epithelial cells [4,5,6].

TSPs are a family of transmembrane proteins which interact with other transmembrane proteins to form a web that stabilizes membranes [7]. Structurally, the tetraspanin protein includes two extracellular loops, a short extracellular loop (SEL) and a large extracellular loop (LEL), both of which decorate the extracellular surface of the cell membrane. TSPs are abundant on the surface of the tegument of the fluke, and EVs, derived from the tegumental surface, are actively internalized by adjacent cholangiocytes, the epithelial cells lining the bile duct [8,9].

Recombinant *Ov*-TSP-2 shows promise as a vaccine candidate in that it confers protection to vaccinated hamsters against challenge infection with *O. viverrini* metacercariae when administered in an adjuvanted form via the parenteral route [4,6]. In addition, oral administration to hamsters of spores of recombinant *Bacillus subtilis*, expressing *Ov*-TSP-2-LEL on the surface of the bacilli, induced serum and bile IgG and IgA responses, and anti-*Ov*-TSP-2 IgG blocked the uptake of *O. viverrini* EVs (*Ov*-EVs) in a human cholangiocyte cell line. Moreover, vaccination provided up to 56% reductions in both worm and egg burdens after challenge infection [10].

To understand the role of antibodies against *Ov*-TSP-2 in conferring protection against *O. viverrini* infection, we raised two monoclonal antibodies (mAbs) to recombinant *Ov*-TSP-2-LEL, both of which significantly blocked the uptake of *Ov*-EVs by human cholangiocytes in vitro, and one of which, after passive transfer to hamsters, conferred highly significant protection against challenge infection with *O. viverrini* compared to a control mAb.

## 2. Materials and Methods

### 2.1. Ethics Statement

Vertebrate animal protocols were approved by the Animal Ethics Committee of Khon Kaen University (approval number ACUC-KKU-121/62) according to the Ethics of Animal Experimentation of the National Research Council of Thailand. Monoclonal antibody production was approved by the James Cook University Animal Care and Use Committee (A2629).

### 2.2. Preparation of Recombinant Ov-TSP-2-LEL Antigen

The recombinant protein corresponding to the large extracellular loop (LEL) of the *Ov*-TSP-2 (GenBank accession JQ678707.1) was produced as a fusion protein with thioredoxin (TRX) using the plasmid pET32a+ (Novagen, Madison, WI, USA), which was expressed by *Escherichia coli* strain BL21DE3. The recombinant fusion protein was purified with a Ni^2+^ affinity column as previously described [8]. The purified protein was dialyzed into PBS before the treatment of mice.

### 2.3. Mouse Immunization

Five-week old male BALB/c mice were immunized a total of 5 times with 50 μL of recombinant *Ov*-TSP-2-LEL at a concentration of 1 mg/mL in an equal volume of Freund’s complete (first immunization) or incomplete (second, third, and fourth immunizations) adjuvants administered subcutaneously at two-week intervals. The fifth (final) immunization was undertaken with 50 μg of recombinant *Ov*-TSP-2-LEL without adjuvant. Serum samples were collected 2 days prior to the third, fourth, and fifth immunizations for the analysis of antibody titers. Mice were housed in the specified pathogen-free animal facility at James Cook University, Cairns, Queensland, Australia.

### 2.4. Hybridoma Generation

Hybridomas were generated by fusing SP2/0 myeloma cells and splenocytes from a vaccinated mouse (Figure 1) 3 days after the final immunization. Briefly, the mouse with the highest antibody titer against recombinant *Ov*-TSP-2-LEL was euthanized and the spleen was immediately removed aseptically and carefully disaggregated using a cell strainer to form a single-cell suspension, followed by 3 washes using a serum-free medium (SFM) (Thermo Fisher Scientific, Waltham, MA, USA). The cells were mixed at a ratio of 1:5 viable parental myeloma cells to each viable splenocyte, and carefully fused using 1 mL of polyethylene glycol (PEG) (Sigma-Aldrich, St. Louis, MO, USA) for 1 min. The fused cells were resuspended in SFM and incubated in a water bath at 37 °C for 15 min, and then centrifuged at 400× *g* for 7 min at room temperature to eliminate the PEG. The supernatant was discarded and the fused cells were gently resuspended in 10 mL of 10% serum containing medium (SCM) (10% of fetal bovine serum in SFM), placed in a T-75 cm^2^ tissue culture flask (Sigma-Aldrich, St. Louis, MO, USA) containing 20 mL of 10% SCM (total culture volume is 30 mL), and incubated at 37 °C and 5% CO_2_ overnight. The fused cell suspension was removed from the flask, centrifuged, and directly transferred to a bottle containing 90 mL of ClonaCell^TM^-HY Medium D (STEMCELL Technologies, Waterbeach, Cambridge, UK) and mixed thoroughly. The Medium D-containing fused cells were carefully transferred to 10 × 100 mm Petri dishes. The dishes containing fused cells were incubated at 37 °C and 5% CO_2_ for 14 days without disturbance. After 14 days, colonies detected on each Petri dish were transferred into an individual well of a 96-well tissue culture plate (Sigma-Aldrich, St. Louis, MO, USA) containing 200 µL of selection media (5% of fetal bovine serum in SFM with 1× hypoxanthine, aminopterin, and thymidine (HAT) (Sigma-Aldrich, St. Louis, MO, USA), and the plates were incubated at 37 °C and 5% CO_2_ for 3–4 days prior to preliminary screening by indirect ELISA.

### 2.5. Expansion and Production of Monoclonal Antibodies

Hybridomas in positive wells of 96-well tissue culture plates were expanded into 24 well-plates and then the supernatant was screened by indirect ELISA, using recombinant *Ov*-TSP-2-LEL and anti-mouse antibodies against IgG (Sigma-Aldrich, St. Louis, MO, USA) and IgM (Sigma-Aldrich, St. Louis, MO, USA). We did not detect any IgG mAbs, however, clones that generated an IgM signal by ELISA were expanded to a T-25 cm^2^ tissue culture flask (Sigma-Aldrich, St. Louis, MO, USA). After several subcloning and screening rounds, the stable, confirmed positive hybridoma cell lines were transferred to T-75 cm^2^ tissue culture flasks (Sigma-Aldrich, St. Louis, MO, USA). For upscaling of hybridoma cells for mAb collection, we used HYPERFlask^®^ M cell culture vessels (High Yield PERformance Flask, Corning Inc., Corning, NY, USA) following the manufacturer’s instructions, with some modification [11]. Briefly, cells in T-75 cm^2^ tissue culture flasks were counted using a hemocytometer. At least 1 × 10^7^ cells in 560 mL of SFM with 1X HAT and 1X Nutridoma-SP (Sigma-Aldrich, St. Louis, MO, USA), were transferred to HYPERFlask^®^ M cell culture vessels (Corning Inc., Corning, NY, USA). After 14 days of incubation, the supernatants were separated from the cells by filtration using a 0.22 μm Stericup^®^-GP quick release sterile vacuum filtration system (Sigma-Aldrich, St. Louis, MO, USA) and transferred to a new 1 L glass bottle prior to the purification of mAbs from the cell supernatant.

### 2.6. Purification of Monoclonal Antibodies

The cell supernatants containing IgM mAbs were purified using a HiTrap^®^ IgM Purification HP (GE Healthcare Bio-Sciences, Uppsala, Sweden) attached to an AKTA^TM^ start chromatography system (GE Healthcare Bio-Sciences, Uppsala, Sweden) following the manufacturer’s instructions as previously described [12]. Briefly, the cell supernatant was precipitated with saturated (NH_4_)_2_SO_4_ to a final concentration of 0.8 M. The ammonium sulfate-containing supernatant was filtered through a 0.45 µm filter immediately before applying it to the column. Before applying the sample to the column, the column was washed with 5 mL of distilled water to remove ethanol, then the column was equilibrated with 5 mL of binding buffer (20 mM sodium phosphate and 0.8 M (NH_4_)_2_SO_4_ with pH 7.5) at a flow rate of 1 mL/min. The sample was applied to the column at a flow rate of 1 mL/min. The unbound sample was washed out using 15 mL of binding buffer with a flow rate of 1 mL/min until the absorbance reached a steady baseline. IgM mAbs were eluted using 12 mL of elution buffer (20 mM sodium phosphate with pH 7.5). After elution, the column was regenerated and washed with 7 mL of wash buffer (20 mM sodium phosphate, pH 7.5 with 30% isopropanol) and re-equilibrated with 5 mL of binding buffer, prior to the subsequent purification, or washed with distilled water then maintained in ethanol for longer term storage.

### 2.7. Screening of Hybridoma Cells by Indirect ELISA

The supernatants of wells and flasks containing hybridomas and purified IgM mAbs were screened for mAb reactivity by indirect ELISA against the recombinant *Ov*-TSP-2-LEL antigen used to immunize the mice, as described previously, with some modification [5]. Briefly, 96-well microtiter plates (Thermo Fisher Scientific, Waltham, MA, USA) were coated with recombinant *Ov*-TSP-2-LEL expressed in *Escherichia coli* and purified using nickel-NTA chromatography [8] at a final concentration of 2 μg/mL overnight at 4 °C. Plates were washed 3 times with PBS 0.05% Tween-20 (PBST) and then blocked with 200 μL of 5% skim milk in PBST for 2 h at room temperature. After washing with PBST, 100 μL of each supernatant or purified IgM mAb (1 mg/mL) was added in duplicate, incubated for 1.5 h at room temperature and then washed with PBST. The plates were probed with 100 μL/well of anti-mouse IgG-HRP (BioRad; diluted 1:5000 in PBST) or anti-mouse IgM μ-chain specific-HRP (Sigma-Aldrich, St. Louis, MO, USA; diluted 1:5000 in PBST) and incubated for 1 h at room temperature. After washing, the plates were developed with 3,3′,5,5′-tetramethylbenzidine (TMB) (Thermo Fisher Scientific, Waltham, MA, USA) and the reaction was stopped with 2 M of H_2_SO_4_. The colorimetric reaction was read at 450 nm on a SpectraMax microplate reader (Molecular Devices, San Jose, CA, USA). Pre-immunization mouse sera and the sera from mice that received the full course of immunization (100 μL, diluted 1:1000 in PBST) were used as negative and positive controls, respectively.

### 2.8. Preparation of O. viverrini Metacercariae

*O. viverrini* metacercariae were prepared as described in [4]. Briefly, cyprinid fishes, from natural sources, were homogenized with an electric blender and then pepsin solution (0.25% pepsin powder, 15% HCl in normal saline solution (NSS)) was added at a ratio of 1:3 volume by volume, followed by incubation at 37 °C for 1 h in a water bath. The digested solution was filtered through 1000, 300, and 106 μm steel sieve meshes. The filtered content obtained by filtering with the 106 μm mesh sieve was washed and repeatedly sedimented with NSS until clear. Sediments were examined for metacercariae under a dissecting microscope. *O. viverrini* metacercariae were collected and stored in sterile NSS at 4 °C until use.

### 2.9. Passive Immunization, Challenge, and Specimen Collection

Fifteen male golden Syrian hamsters (*Mesocricetus auratus*) 6–8 weeks of age, reared at the animal facility of the Faculty of Medicine, Khon Kaen University, were randomly divided into three groups. Groups 1 and 2 received intraperitoneal administration of the anti-*Ov*-TSP-2 mAbs designated 1D6 and 3F5. Group 3 was vaccinated with a commercially purchased mouse monoclonal IgM isotype control (Thermo Fisher Scientific, Waltham, MA, USA). The time course of passive immunization, fluke challenge infection, and specimen collection is shown in Figure 2. The hamsters were intraperitoneally immunized with a single 200 μg dose of 1D6, 3F5, or control IgM in 500 μL of dH_2_O one day before challenge (day −1). The following day (day 0), all immunized hamsters (*n* = 5 per group) were challenged with 50 *O. viverrini* metacercariae through orogastric administration. Eight weeks later, all hamsters were euthanized. At necropsy, their livers were collected for worm counts. The sera of hamsters were collected three times at pre-immunization (day −1), post-immunization/pre-challenge (day 0), and post-challenge at necropsy (week 8 post-infection).

### 2.10. Fecal Egg Counts and Worm Recovery

Hamster feces were collected at week 7, after the challenge with *O. viverrini* metacercariae, for egg counts. A modified formalin-ether acetate concentration technique (FECT) was used to determine the number of eggs per gram of feces (EPG). Whole livers were removed from hamsters at necropsy, dissected in NSS, and adult flukes were gently removed by squeezing the tissue to obtain the flukes from the bile ducts to determine adult worm burdens. To measure worm length, a total of 40, 30, and 20 worms from the control IgM group, 1D6, and 3F5, respectively (less worms were recovered from the two test groups), were randomly selected, washed three times with NSS and fixed in pre-warmed 10% formalin. The worms were photographed under microscopy and the worm length was measured using NIS Element software (Nikon, Japan).

### 2.11. Detection of Hamster Anti-*Ov*-TSP-2-LEL-Specific IgG, IgM, and IgA

Hamster anti-*Ov*-TSP-2-LEL-specific IgG, IgM, and IgA in sera were measured by ELISA, as described, with some modification, in [10]. Briefly, 96-well microtiter plates (Thermo Fisher Scientific, Waltham, MA, USA) were coated with 100 μL of recombinant *Ov*-TSP-2-LEL expressed in *E. coli* and purified using nickel-NTA chromatography [8] (2 μg/mL) overnight at 4 °C in 0.05 M of Na_2_CO_3_/NaHCO_3_ with a pH of 9.6 (coating buffer). The plates were washed and blocked as described above. One hundred μL of sera (1:500 in PBST and 2% skim milk) was added and incubated for 1.5 h at room temperature. The plates were washed with PBST and probed with 100 μL of anti-hamster IgG-HRP (BioRad, Hercules, CA, USA; diluted 1:2000 in PBST), HRP-conjugated anti-IgM (μ-chain specific) (Sigma-Aldrich, St. Louis, MO, USA, diluted 1:2000 in PBST), and anti-IgA-HRP (Invitrogen, Carlsbad, CA, US; diluted 1:500 in PBST) was then added and incubated for 1 h at room temperature. After, the washing plates were developed using TMB. Pre-immunization hamster sera and the sera from hamsters that received the full course of oral immunization of recombinant *Bacillus subtilis* spores expressing *Ov*-TSP-2-LEL, as described previously [10] (100 μL, diluted 1:1000 in PBST), served as negative and positive controls, respectively.

### 2.12. O. viverrini EV Internalization by Human Biliary Epithelial Cells

A total of 1.25 μg (5 × 10^7^ vesicles) of *O. viverrini* EVs were labeled with PKH67 (Sigma-Aldrich, St. Louis, MO, USA), following the manufacturer’s instructions as described previously [4], and incubated with pooled pre-vaccination serum at a dilution of 1:2.5 and 1.25 μg of 1D6, 3F5, or the control IgM at a dilution of 1:1 for 1 h at room temperature with periodic mixing [9]. After incubation, EV-antibody complexes were washed with PBS using Amicon Ultra 100 kDa cut-off purification columns (Merck Millipore, Burlington, MA, USA) and cultured with the H69 normal human biliary cell line [13] at 37 °C with 5% CO_2_ for 2 h. The nuclei were stained with 2 μg/mL of Hoechst (Invitrogen, Carlsbad, CA, USA) for 15 min at room temperature. Fluorescence images were captured by confocal microscopy (Carl Zeiss LSM800, Dublin, CA, USA) at 200× original magnification. Thirty cells from two biological replicates were analyzed for fluorescence intensity using ImageJ version 1.52a.

### 2.13. Statistical Analysis

Experimental values were expressed as mean ± standard deviation (SD). The data were analyzed using one-way analysis of variance (ANOVA) and two-way ANOVA using GraphPad Prism 9 software version 9.1.1 (GraphPad Software Inc., San Diego, CA, USA), and an unpaired t-test was employed to compare the two normally distributed groups. A *p* values of ≤ 0.05 was considered to be statistically significant.

## 3. Results

### 3.1. Generation and Characterization of Anti-Ov-TSP-2-LEL mAbs

Following the fusion of SP2/0 myeloma cells with spleen lymphocytes from mice vaccinated with recombinant *Ov*-TSP-2-LEL, the fused cells were isolated and cloned using the semi-solid medium ClonaCell^TM^-HY Medium D. After incubation for 14 days at 37 °C, the hybridoma cells were harvested from the semi-solid medium. At least 250 colonies were observed in Petri dishes and removed from the plate by transferring each colony into an individual well of a 96-well tissue culture plate containing a medium. The supernatant of each hybridoma was screened for reactive antibodies against recombinant *Ov*-TSP-2-LEL using indirect ELISA. From ≥250 colonies, four mAb-producing hybridomas generated a positive ELISA signal in the preliminary screen. Positive clones from the 96-well tissue culture plate were expanded by transferring cells to a 24-well plate, and after expansion, two hybridomas (1D6 and 3F5) were selected due to their strong reactivity against recombinant *Ov*-TSP-2-LEL. Isotyping of the hybridomas, using IgG (Fc)- and IgM-specific-secondary antibodies, revealed that these two mAb-producing hybridomas were IgM. 1D6 and 3F5 were expanded by transferring 1 mL of cells from a 24-well plate to a T-25 cm^2^ tissue culture flask containing 9 mL of 5% SCM. After several rounds of subcloning and screening, each stable hybridoma cell line was transferred to a T-75 cm^2^ tissue culture flask. For large scale production of hybridomas producing anti-*Ov*-TSP-2-LEL IgM mAb, the hybridomas were transferred to HYPERFlask^®^ M cell culture vessels. After 14 days of incubation, the supernatant was separated from the cell suspension and the mAbs were purified using HiTrap^®^ IgM Purification HP. At least 1 mg of each anti-*Ov*-TSP-2-LEL IgM mAb, clones 1D6 and 3F5, was purified for downstream use in passive immunizations.

### 3.2. Passive Immunization with Anti-Ov-TSP-2-LEL IgM mAbs and Detection in Serum

Sera were collected from the hamsters pre-immunization, pre-challenge, and post-challenge to identify mAbs in circulation and to quantify the specific antibody responses generated against *Ov*-TSP-2-LEL following parasite challenge. The hamsters immunized with 1D6 and 3F5 had significantly higher serum IgG levels against *Ov*-TSP-2-LEL post-challenge compared to pre-immunization and pre-challenge (*p* < 0.05 and *p* < 0.01, respectively). In addition, and as expected, serum IgG levels against *Ov*-TSP-2-LEL post-challenge were not significantly different when compared among all three groups (Figure 3A).

Hamsters immunized with 1D6 and 3F5 had significantly higher serum IgM levels against *Ov*-TSP-2-LEL post-immunization compared to pre-immunization (*p* < 0.05 and *p* < 0.01, respectively). In addition, post-immunization anti-*Ov*-TSP-2-LEL serum IgM levels of hamsters immunized with 1D6 and 3F5 were significantly higher than IgM levels of hamsters immunized with the control IgM (*p* < 0.05). Hamsters immunized with the control IgM had significantly higher serum IgM levels against *Ov*-TSP-2-LEL post-challenge compared to pre-immunization and pre-challenge (*p* < 0.0001) as a result of the immunogenicity of TSP-2 in natural infections. Hamsters immunized with 1D6 had significantly higher serum IgM levels against *Ov*-TSP-2-LEL post-challenge compared to both pre-immunization and pre-challenge (*p* < 0.0001 and *p* < 0.001, respectively). Hamsters immunized with 3F5 had significantly higher serum IgM levels specific to *Ov*-TSP-2-LEL post-challenge compared to pre-immunization (*p* < 0.0001), but this did not reach significance when compared to pre-challenge levels (Figure 3B).

### 3.3. Passive Immunization with Anti-Ov-TSP-2-LEL IgM mAbs Induced Partial Protection Against O. viverrini Infection

Hamsters immunized with 1D6 and 3F5 had a 19.8% and 74.5% reduction in worm burdens, respectively, compared to hamsters immunized with the control IgM. The average worm burden from 3F5-immunized hamsters was 5.4 ± 2.97, while from 1D6-immunized hamsters it was 17 ± 3.08, and from control IgM-immunized hamsters it was 21.2 ± 4.92. The worm burdens of hamsters vaccinated with 3F5 were significantly lower than those from hamsters that received 1D6 (*p* < 0.01) and the control IgM (*p* < 0.0001). No significant differences were observed in worm burdens between hamsters that received 1D6 and the control IgM (Figure 4A).

The fecal EPG levels of hamsters immunized with 3F5 were significantly lower than that of hamsters that received the control IgM, 682 ± 372 (mean ± SD) vs. 2089 ± 389, respectively. This equates to a 67.4% reduction (*p* < 0.001). No significant differences were observed between the fecal EPG levels from hamsters that received 1D6 (1920 ± 560) or the control IgM (Figure 4B).

The mean length of examined worms from hamsters passively immunized with 1D6, 3F5, and the control IgM mAbs was 4.0 ± 0.68 mm, 3.9 ± 0.73 mm, and 4.29 ± 0.89 mm, respectively (Figure 4C; not statistically significant).

### 3.4. Antibodies from Passively Immunized Hamsters Block Uptake of O. viverrini EVs by Human Cholangiocytes

PKH67-labelled *O. viverrini* EVs were incubated with pooled pre-vaccination serum at a dilution of 1:2.5, or 1.25 μg of the three mAbs (at a dilution of 1:1), before being cultured with H69 cholangiocytes. The uptake of EVs was significantly reduced by 1D6 (97.5% reduction) and 3F5 (97% reduction) compared to the hamster pooled pre-vaccination serum. The uptake of EVs was also significantly reduced by 1D6 (97.9% reduction) and 3F5 (97.4% reduction) compared to the control IgM (Figure 5).

## 4. Discussion

Here, we showed that hamsters passively immunized with an anti-*Ov*-TSP-2-LEL IgM mAb had reduced worm and egg burdens after challenge with *O. viverrini* metacercariae. Circulating mAb was readily detected in the sera of vaccinated hamsters one day post-intraperitoneal immunization, and we hypothesize that transferred antibodies readily bound to parasite tegument and secreted EVs expressing *Ov*-TSP-2 as the worms migrated through the tissues en route to the bile ducts. These findings indicate that *Ov*-TSP-2-LEL is a promising vaccine antigen, which is consistent with findings in our earlier studies that showed the efficiency of this tetraspanin fragment as a component of a subunit vaccine in both parenteral and oral form [4,6,10].

Both 1D6 and 3F5 mAbs almost completely blocked the in vitro uptake of *O. viverrini* EVs by cholangiocytes, but only 3F5 provided significant protection against challenge infection. Since we did not assess increasing quantities of the mAbs by passive transfer followed by challenge infection, we cannot exclude that 1D6 may confer protection at a higher concentration than that tested here. It is also unclear whether the two mAbs bind to different epitopes. We did assess the binding of the mAbs by ELISA to overlapping 13-mer synthesized peptides corresponding to *Ov*-TSP-2 LEL, but the data were unconvincing, possibly due to the reduced specificity of antigen binding displayed by IgM antibodies (not shown).

At least 570 therapeutic mAbs were studied in clinical trials [14], and 79 therapeutic mAbs were approved by the United States Food and Drug Administration and are currently on the market, including 30 mAbs for the treatment of cancer [15]. However, whereas therapeutic mAbs against bacterial and viral infections were approved, there are no antibody-based therapies in use for parasitic infections [16,17].

There are five antibody isotypes in mice and humans: IgG, IgM, IgA, IgE, and IgD. IgM antibodies are either pentameric or hexameric, found in blood, and function similarly to IgG in defending against antigens. IgM is the main antibody produced in an initial attack by a specific bacterial or viral antigen, while IgG is usually produced in later infections caused by the same agent. IgM antibodies were shown to inhibit infectivity of organisms by causing aggregation or agglutination of the pathogen or infected cell, leading to its clearance from the body [18]. IgM plays an essential role in protective immune responses against numerous parasitic helminths. For example, in a mouse model of human strongyloidiasis, immunization with infective larvae induced protective immunity that could be passively transferred to naïve mice [19]. Parasite-specific IgG1, IgM, and IgA titers were elevated in immunized mice, but IgM was the only isotype capable of inducing parasite elimination, and this was dependent on cell contact, the presence of granulocytes, and complement activation. Moreover, IgM from immunized mice passively transferred immunity to naïve IL-5 knockout mice, which are deficient in eosinophils [20], suggesting that neutrophils were the required granulocyte for IgM-dependent killing [21]. Moreover, an IgM mAb that targeted multiple antigens from the filarial nematode *Brugia malayi*, conferred 89% protection when passively transferred to jirds that were challenged with infective larvae, and was thought to function in an antibody-dependent cell-mediated cytotoxicity (ADCC) and complement-dependent manner [22].

While IgM mAbs that confer passive protection against liver flukes have not (to the best of our knowledge) been described, they were for other platyhelminth infections, primarily schistosomiasis. Two IgM mAbs against *Schistosoma mansoni* 28 kDa glutathione S-transferase (GST), one of which had enzyme neutralizing properties, conferred protection against challenge infection when transferred to rats [23]. Eight mAbs were raised to surface antigens of *Schistosoma japonicum*, only one of which was an IgM that conferred protection via passive transfer studies in mice [24]. An IgG mAb was raised to 78 kDa ES protein-protected mice against challenge infection with the livestock liver fluke *Fasciola hepatica* [25], but to the best of our knowledge, mAbs raised against human liver fluke antigens have only been used to develop diagnostic tools [26,27,28,29] and were not shown to confer protective efficacy in passive transfer studies.

We found that both serum IgG and IgM levels increased post-challenge in hamsters that received both 3F5 and 1D6 as well as control IgM, indicating that these responses were a result of parasite challenge and not passive transfer of mAb. Due to the fact we passively immunized hamsters with anti-*Ov*-TSP-2-LEL IgM mAbs, antigen-specific IgG responses were not detectable pre-challenge and were only observed post-challenge as a result of infection. However, as expected, we detected increased antigen-specific IgM levels in serum pre-challenge from hamsters that received 3F5 and 1D6, but not the control IgM. The *Ov*-TSP-2-specific IgM levels increased again to comparable levels in all three groups post-challenge (including the control IgM group), implying that challenge infection promoted a strong anti-TSP-2 IgM response in all groups.

Both 3F5 and 1D6 mAbs almost completely blocked the uptake of fluke EVs by cholangiocytes in vitro, whereas control IgM had no effect, providing a plausible mechanism by which the vaccination strategy exerted its effect. Others have shown that anti-*O. viverrini* IgG, IgM, and IgA were found at significantly higher levels in stools of egg-negative, compared to egg positive, residents in *O viverrini*-endemic areas, suggesting that all antibody isotypes contribute to naturally occurring protective responses against opisthorchiasis [30].

## 5. Conclusions

We showed that passive immunization of hamsters with an anti-*Ov*-TSP-2-LEL IgM mAb elicited protection against *O. viverrini* infection in hamsters, possibly via binding to, and inhibition of, EV uptake by host biliary epithelial cells, thereby interrupting vesicle-mediated host-parasite communication. This study lends further support to the efficacy of antibodies against EV and tegument tetraspanins as a basis for vaccine development against carcinogenic human liver fluke infection. Anti-TSP-2 mAbs also have a potential role as diagnostic reagents for opisthorchiasis given the abundance of *Ov*-TSP-2 on EVs, particularly in light of the recent report on *S. mansoni* EVs as serum biomarkers of schistosomiasis [31].

Future work will explore the mechanism by which anti-TSP-2 antibodies exert their efficacy and whether ADCC and complement play essential roles in parasite killing. Moreover, mapping of the protective epitopes within the LEL of *Ov*-TSP-2 will facilitate the design of a chimeric vaccine construct that targets numerous antigens and parasitism pathways to obtain the greatest possible efficacy.

## Figures and Tables

**Figure 1 vaccines-09-00740-f001:**
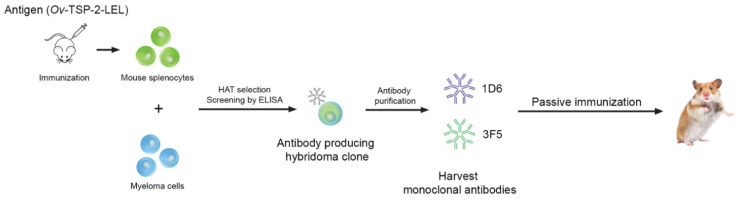
Hybridoma generation was performed by immunization of mice with adjuvanted *Ov*-TSP-2-LEL and fusion of splenocytes with SP2/0 myeloma cells. After isolation of hybridomas by HAT selection, the supernatants that contained mAbs from each clone were screened by ELISA to select antibody producing clones. After several rounds of screening, two IgM antibody producing clones, 1D6 and 3F5, were selected and purified for passive immunization of hamsters.

**Figure 2 vaccines-09-00740-f002:**
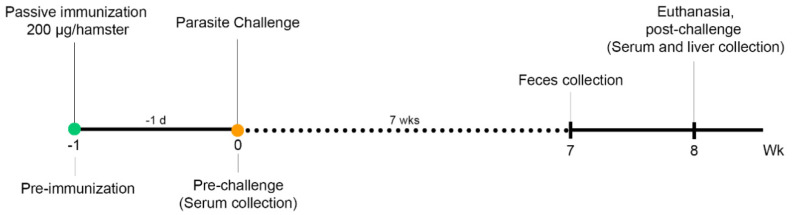
Schematic representation of the hamster vaccination and challenge regimen.

**Figure 3 vaccines-09-00740-f003:**
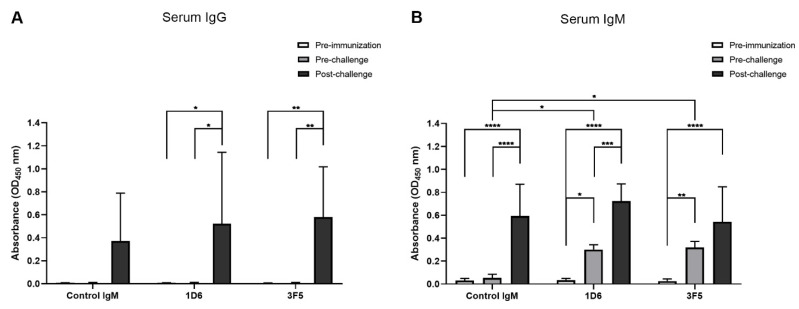
Anti-*Ov*-TSP-2 serum IgG and IgM levels in passively immunized hamsters determined by ELISA at pre-immunization (day -1), post-immunization/pre-challenge (day 0), and post-challenge at necropsy (week 8 post-infection). (**A**) Serum IgG and (**B**) serum IgM against recombinant *Ov*-TSP-2-LEL was measured from hamsters that were passively immunized with 1D6, 3F5, and the IgM isotype control. Results represent the mean absorbance at 450 nm for each group. * *p* < 0.05, ** *p* < 0.01, *** *p* < 0.001, and **** *p* < 0.0001.

**Figure 4 vaccines-09-00740-f004:**
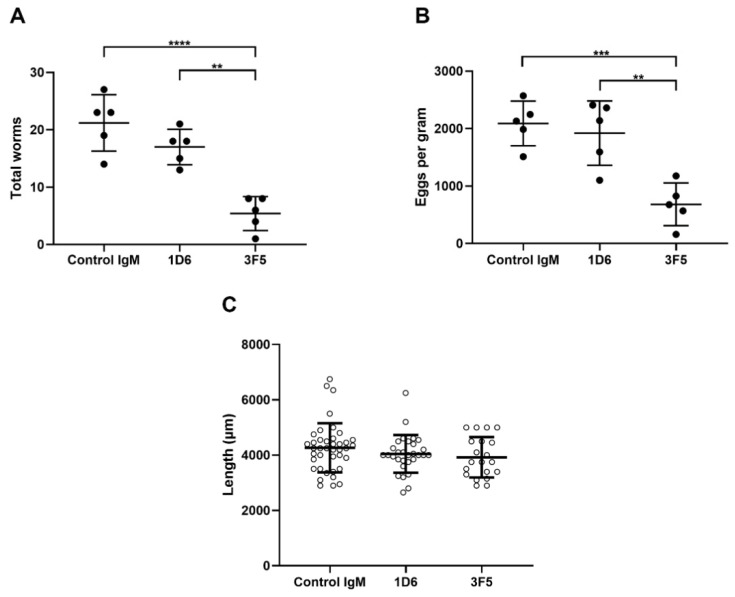
(**A**) Hamsters immunized with 3F5 harbored significantly fewer worms and (**B**) passed significantly fewer fecal eggs than hamsters immunized with 1D6 or isotype control mAb. (**C**) The average lengths of liver flukes recovered from hamsters that were immunized with 1D6 and 3F5 showed a trend towards stunting (4.0 ± 0.68 mm and 3.9 ± 0.73 mm, respectively) when compared to the isotype control group (4.29 ± 0.89 mm), but the differences did not reach significance (*p* = 0.48 and *p* = 0.26, respectively). ** *p* < 0.01, *** *p* < 0.001, and **** *p* < 0.0001.

**Figure 5 vaccines-09-00740-f005:**
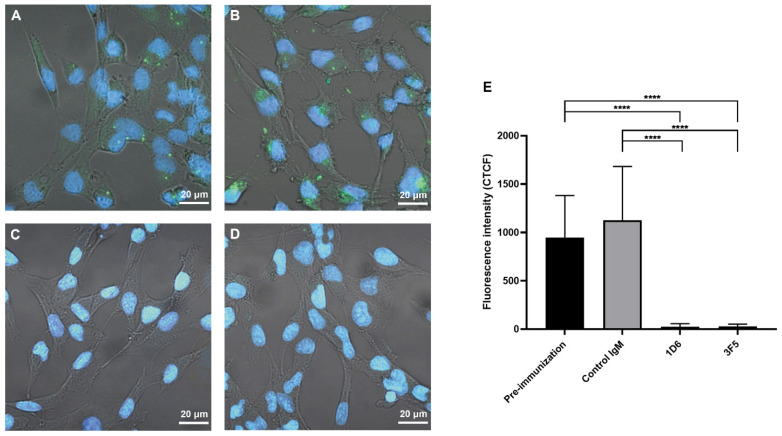
Anti-*Ov*-TSP-2-LEL mAbs (1D6 and 3F5) block *Opisthorchis viverrini* EV internalization by the H69 human cholangiocyte cell line. A total of 1.25 μg (5 × 10^7^ vesicles) of PKH67-labeled *O. viverrini* extracellular vesicles were incubated with (**A**) hamster pooled pre-vaccination serum (at a dilution of 1:2.5) and incubated with a 1.25 μg of the three mAbs including (**B**) mouse IgM isotype control mAb, (**C**) anti-*Ov*-TSP-2-LEL mAbs 1D6, and (**D**) anti-*Ov*-TSP-2-LEL mAbs 3F5, before co-culture with H69 cholangiocytes (15,000 cells per well) for 2 h. (**E**) Corrected total cell fluorescence (CTCF) at 490 nm (green channel) was quantified using ImageJ software. The nuclei were stained blue using Hoechst dye no. 33258. Bars indicate standard deviation of the mean. **** *p* < 0.0001.

## Data Availability

Data are contained within the article.

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
