# Peer review of "Monoclonal Antibodies Targeting an Opisthorchis viverrini Extracellular Vesicle Tetraspanin Protect Hamsters against Challenge Infection"

_vaccines, 2021, doi:10.3390/vaccines9070740_

Round 1
Reviewer 1 Report
Phumrattanaprapin et all provide a nice followup to their previous findings that immunization against the Opisthorchis viverrini Tetraspanin found in extracellular vesicles can provide immunity against infection. Now they have taken the antibodies they have found to be produced upon immunization and use them as a passive immune therapy against Opisthorchis viverrini. Their challenges against hamsters that were passively immunized show promise in lower parasite burden and provide an avenue for further development of parasitic therapies. The importance of these types of passive therapies in the future of travel medicine as well as therapies endemic with particular parasites is not lost and I encourage this work going forward.
One small note is that Figure 3 should reference Figure 2 in the legend to help the reader.
Author Response
Reviewer #1:
Phumrattanaprapin et al provide a nice follow-up to their previous findings that immunization against the Opisthorchis viverrini Tetraspanin found in extracellular vesicles can provide immunity against infection. Now they have taken the antibodies they have found to be produced upon immunization and use them as a passive immune therapy against Opisthorchis viverrini. Their challenges against hamsters that were passively immunized show promise in lower parasite burden and provide an avenue for further development of parasitic therapies. The importance of these types of passive therapies in the future of travel medicine as well as therapies endemic with particular parasites is not lost and I encourage this work going forward.
One small note is that Figure 3 should reference Figure 2 in the legend to help the reader.
- Timeline of serum collection from Figure 2 has been added to Figure 3.
Reviewer 2 Report
This paper by W. Phumrattanaprapin et al. shows that a monoclonal IgM (not IgG) raised against the Tetraspanin (TSP) protein of Opisthorchis viverrini, the human liver fluke, is effective in inhibiting infection in the hamster model. The experiments are well done and the results are promising for passive immunization against the flute worm, which establishes chronic hepatobiliary infection, primarily in areas where ingestion of uncooked fish is common. The use of the soluble extracellular loop regions of this parasitic membrane protein as antigen in the hybridomas was prudent, although the exact peptide epitope of the IgM’s were not mapped. The generation of the monoclonal(s) and their test for protection against the parasite involve procedures that are well established, offering confidence in the work.
Nonetheless, I have several comments and suggestions, as listed below.
(1) The authors should at least provide a speculative hypothesis why the two independent anti-TSP monoclonals were both IgM and not IgG.
(2) Last paragraph of Introduction (Line 61-68) should be rewritten because it is self-defeating as presented. It is currently stated that this research started with the goal of identifying multiple antigens “in addition to TSPs” targeting parasitism. But the paper used TSP only! Moreover, at the end, it did not actually identify the epitopes against which the two monoclonals were produced. So, we did not even know if they are entirely different or overlapping.
(3) Hamster pathology: There is no statement on the physiology of the hamsters, i.e. their pathology and its relief in the monoclonal-treated animals (at least for the anti-3F5, the better acting IgM) over the control-IgM animals. Were the animals followed long-term to see if chronic infection was established or not?
(4) Prior failures or future prospects? The paragraph of Line 394-403 trumpets prior success of IgG’s in the clinic, in contrast to that of the IgM’s of IgA’s. While this sounds sobering for the clinical prospects of the current IgM, the next two paragraphs offer hope for anti-parasite IgM’s. Perhaps some connecting sentences are needed to offer a better flow between them.
(5) Section 3.2.” “Passive immunization… Detection in Serum”: What is the point here? If the animals are given IgM, isn’t it expected that the IgM will appear in their system. Why was IgG even assayed? A rationale for these studies and a concluding line would help.
Minor language recommendations in the Abstract
Line 22: Put a hyphen in “opisthorchiasis induced”.
Line 26: Delete “by generating mAbs specific to Ov-TSP-2”. As written, it is superfluous, since it goes without saying that in order to “assess the protective capacity" of a antibody, the antibody must be generated first!
Line 30-31: Insert “the” between “This is” and “first report”.
Author Response
Reviewer #2:
This paper by W. Phumrattanaprapin et al. shows that a monoclonal IgM (not IgG) raised against the Tetraspanin (TSP) protein of Opisthorchis viverrini, the human liver fluke, is effective in inhibiting infection in the hamster model. The experiments are well done and the results are promising for passive immunization against the flute worm, which establishes chronic hepatobiliary infection, primarily in areas where ingestion of uncooked fish is common. The use of the soluble extracellular loop regions of this parasitic membrane protein as antigen in the hybridomas was prudent, although the exact peptide epitope of the IgM’s were not mapped. The generation of the monoclonal(s) and their test for protection against the parasite involve procedures that are well established, offering confidence in the work.
Nonetheless, I have several comments and suggestions, as listed below.
(1) The authors should at least provide a speculative hypothesis why the two independent anti-TSP monoclonals were both IgM and not IgG.
We performed 6 injections over a twelve-week period, which should have ensured class switching and promoted IgG hybridoma formation. Nonetheless, we only detected IgM secreting mAbs. We do not have a logical explanation for the absence of IgG secreting hybridomas. It might be a function of the specific protein immunogen. Despite our expectations of generating IgG mAbs, we decided to proceed with IgM mAbs given the polyvalency of IgM and associated high avidity binding and enhanced engagement of complement compared to IgG.
(2) Last paragraph of Introduction (Line 61-68) should be rewritten because it is self-defeating as presented. It is currently stated that this research started with the goal of identifying multiple antigens “in addition to TSPs” targeting parasitism. But the paper used TSP only! Moreover, at the end, it did not actually identify the epitopes against which the two monoclonals were produced. So, we did not even know if they are entirely different or overlapping.
We thank the reviewer for highlighting this over sight, and we revised the last paragraph of the Introduction to emphasize the goal of assessing the role of anti-TSP-2 antibodies in conferring protection.
(3) Hamster pathology: There is no statement on the physiology of the hamsters, i.e. their pathology and its relief in the monoclonal-treated animals (at least for the anti-3F5, the better acting IgM) over the control-IgM animals. Were the animals followed long-term to see if chronic infection was established or not?
Thank you very much for the suggestion. We have not investigated pathology changes in this study but we assume that liver pathology would be lessened in the mAb-treated animals, especially the group that received 3F5 due to the reduced numbers of flukes. We plan to investigate this aspect of immunopathology in the future, including the impact of mAb therapy on protecting against the development of fluke-induced cholangiocarcinoma. Of course these studies require vaccinated animals to be kept alive for at least 6 months post-parasite challenge to allow chronic infection to induce cancer.
(4) Prior failures or future prospects? The paragraph of Line 394-403 trumpets prior success of IgG’s in the clinic, in contrast to that of the IgM’s of IgA’s. While this sounds sobering for the clinical prospects of the current IgM, the next two paragraphs offer hope for anti-parasite IgM’s. Perhaps some connecting sentences are needed to offer a better flow between them.
We thank the reviewer for pointing out the poor flow between these sentences. Upon reflection, we decided to delete the sentences about clinical use of IgM mAbs and now the paragraph focuses on the role of IgM only. We believe the flow is now improved and the important content has been retained.
(5) Section 3.2.” “Passive immunization… Detection in Serum”: What is the point here? If the animals are given IgM, isn’t it expected that the IgM will appear in their system. Why was IgG even assayed? A rationale for these studies and a concluding line would help.
The point of assessing both isotypes was to show that transferred IgM could be detected in the serum and that both TSP-2-specific IgM and IgG responses could be detected against challenge infection due to the inherent immunogenicity of the TSP-2 protein during fluke infection. The sentences have been modified to make this differentiation clearer.
Minor language recommendations in the Abstract
Line 22: Put a hyphen in “opisthorchiasis induced”.
- Fixed
Line 26: Delete “by generating mAbs specific to Ov-TSP-2”. As written, it is superfluous, since it goes without saying that in order to “assess the protective capacity" of a antibody, the antibody must be generated first!
- Amended
Line 30-31: Insert “the” between “This is” and “first report”.
- Fixed
